# The anticancer natural product ophiobolin A induces cytotoxicity by covalent modification of phosphatidylethanolamine

Christopher Chidley[1,2,3,4], Sunia A Trauger[5], Kıvanç Birsoy[6], Erin K O'Shea[1,2,3,4]*

[1]Faculty of Arts and Sciences Center for Systems Biology, Harvard University, Cambridge, United States; [2]Department of Molecular and Cellular Biology, Harvard University, Cambridge, United States; [3]Department of Chemistry and Chemical Biology, Harvard University, Cambridge, United States; [4]Howard Hughes Medical Institute, Harvard University, Cambridge, United States; [5]Small Molecule Mass Spectrometry Facility, Faculty of Arts and Sciences Center for Systems Biology , Harvard University, Cambridge, United States; [6]Laboratory of Metabolic Regulation and Genetics, Rockefeller University, New York, United States

**Abstract** Phenotypic screens allow the identification of small molecules with promising anticancer activity, but the difficulty in characterizing the mechanism of action of these compounds in human cells often undermines their value as drug leads. Here, we used a loss-of-function genetic screen in human haploid KBM7 cells to discover the mechanism of action of the anticancer natural product ophiobolin A (OPA). We found that genetic inactivation of de novo synthesis of phosphatidylethanolamine (PE) mitigates OPA cytotoxicity by reducing cellular PE levels. OPA reacts with the ethanolamine head group of PE in human cells to form pyrrole-containing covalent cytotoxic adducts and these adducts lead to lipid bilayer destabilization. Our characterization of this unusual cytotoxicity mechanism, made possible by unbiased genetic screening in human cells, suggests that the selective antitumor activity displayed by OPA may be due to altered membrane PE levels in cancer cells.

*For correspondence: osheae@ hhmi.org

## Introduction

Natural products are an important source for the development of pharmaceutical drugs, especially in oncology; half of all anticancer drugs developed since the 1940s are natural products or derivatives of natural products (*Newman and Cragg, 2012*). Compounds with anticancer activity can be readily identified in cytotoxicity assays and other phenotypic screens (*Harvey et al., 2015*; *Eggert, 2013*). To use these small molecules as anticancer drug leads or to identify new chemotherapy molecular targets, it is essential to characterize the molecular mechanism of action (MOA) that underlies cytotoxicity (*Schenone et al., 2013*; *Bunnage et al., 2015*). However, as unraveling the MOA of bioactive small molecules remains challenging and time consuming (*Schenone et al., 2013*; *Ziegler et al., 2013*), the MOA of many natural products that display promising anticancer activities in phenotypic screens remains uncharacterized (*Shoemaker, 2006*). An example of such a natural product is ophiobolin A (OPA), a plant toxin isolated from pathogenic fungi of the *Bipolaris* genus which displays cytotoxicity at nanomolar concentrations against a range of cancer cell lines (*Au et al., 2000*; *Bury et al., 2013*). OPA induces paraptosis, a form of non-apoptotic cell death, in glioblastoma cells and displays antitumor activity in a mouse glioblastoma model (*Bury et al., 2013*; *Dasari et al., 2015*). The toxicity of OPA to plants is believed to involve calmodulin inhibition via formation of a covalent adduct between OPA and specific lysine side

**eLife digest** Many of the medications that are available to treat cancer are either collected from natural sources or inspired by molecules existing in nature. While it is often challenging to understand how these natural compounds selectively kill cancer cells, characterizing these mechanisms is essential if researchers are to develop new anticancer drugs and treatments based on these compounds.

Ophiobolin A is a compound naturally made by a fungus in order to attack plant cells. It is also able to potently kill cancer cells from humans. In particular, ophiobolin A is a promising candidate for treatment of a type of brain tumor called glioblastomas, which are notoriously difficult to treat with existing medications.

Using a newly developed method, Chidley et al. have now tested which components of human cancer cells are important for ophiobolin A to exert its killing effect. The method revealed that ophiobolin A was less able to kill cancer cells if the cells had lower levels of a molecule called phosphatidylethanolamine in their surface membranes. This observation led Chidley et al. to show that ophiobolin A enters the membrane of human cancer cells and combines chemically with phosphatidylethanolamine to form a new composite molecule. Further experiments showed that the formation of this composite molecule disrupted a model membrane, which suggests that ophiobolin A kills cancer cells by breaking their membranes.

The next challenge is to understand exactly how the composite molecule kills cancer cells via membrane disruption. It also remains unclear if the anticancer activity of ophiobolin A results from cancer cells having a membrane composition that is different from normal cells, and why this difference arises in the first place.

chains (Leung et al., 1984). More recently, it has been shown in synthetic studies that primary amines react with the 1,4-dicarbonyl moiety of OPA to form covalent adducts and that this moiety is critical for animal cell cytotoxicity, leading the authors to suggest that the MOA of OPA in animal cells is through covalent modification of an unknown intracellular target protein (Dasari et al., 2015). In conclusion, OPA represents an interesting candidate for the treatment of glioblastomas that are resistant to classical pro-apoptotic therapeutic approaches, but the lack of information on cellular targets of OPA impedes any further development.

Genetic screens represent an unbiased genome-wide approach to identify molecular targets involved in small molecule MOA but have been mainly limited to application in genetically tractable organisms such as *Saccharomyces cerevisiae* (Roemer et al., 2012; Lee et al., 2014). Recent technical breakthroughs, such as insertional mutagenesis in haploid cells (Carette et al., 2009, 2011) and CRISPR-Cas9 genome editing (Wang et al., 2014; Shalem et al., 2014; Gilbert et al., 2014; Smurnyy et al., 2014) have revolutionized the use of genetic screens in human cell lines to facilitate the study of the MOA of bioactive molecules in model systems more relevant to human disease (Nijman, 2015).

To unravel the MOA of OPA, we took advantage of a genome-wide strategy in human cells to identify genes that are required for OPA to exert its cytotoxic effect. We used insertional mutagenesis in the near-haploid human cell line KBM7 to generate loss-of-function mutants and then selected for growth of cell lines resistant to OPA treatment (Carette et al., 2011). We discovered that inactivation of the pathway for de novo synthesis of phosphatidylethanolamine (PE), also named the Kennedy pathway, confers resistance to OPA. Increased OPA resistance was correlated with decreased cellular PE levels. Surprisingly, we determined that the molecular target of OPA is PE itself; OPA forms a covalent adduct with PE in human cells. This work illustrates the power of unbiased genetic screens in human cells in discovering novel MOAs of compounds identified in phenotypic screens.

## Results

### Identification of targets of anticancer drugs in human cells

Loss-of-function genetic screens in human KBM7 cells have been used to identify cellular factors that are necessary for entry of viruses and bacterial toxins, and transporters of small molecules, but rarely to identify the molecular targets of small molecule drugs (*Carette et al., 2011*; *Nijman, 2015*; *Winter et al., 2014*; *Reiling et al., 2011*; *Birsoy et al., 2013*). Before applying the screen to studies of compounds of unknown MOA displaying anticancer activity, we first determined whether such screens can robustly identify genes involved in the cytotoxicity of anticancer drugs with well-characterized MOAs. Briefly, we used a retroviral gene-trap approach to generate approximately 75 million insertions in the near-haploid human cell line KBM7, covering more than 95% of all expressed genes (*Carette et al., 2011*). This library of loss-of-function cell lines was treated with a toxic dose of anticancer drug and resistant mutants were allowed to grow over 3 weeks. Retroviral insertion sites were identified by amplification of the genomic sequence flanking the insertion site, high-throughput sequencing, and mapping to the human genome. Insertions in exonic regions or in the sense orientation of intronic regions are typically expected to cause gene inactivation (*Carette et al., 2009*). For each gene locus, we calculated an enrichment p-value by comparing the number of inactivating insertions in the pooled drug-resistant cells to the number of such insertions in mutagenized cells before selection. This enrichment p-value allows the identification of genes whose inactivation renders cells resistant to the toxic effects of the small molecule tested (*Figure 1—figure supplement 1*).

We performed screens with anticancer drugs including topoisomerase inhibitors (topotecan, etoposide, and doxorubicin), a proteasome inhibitor (bortezomib), an antimetabolite (gemcitabine) and a platinum-based DNA crosslinking agent (oxaliplatin). As expected, we observed enrichment of inactivating insertions in genes known to play a role in the MOA of these anticancer compounds (*Figure 1—figure supplement 2*). For instance, doxorubicin and etoposide induce cytotoxicity by forming a ternary complex with DNA and the enzyme topoisomerase IIA (*Pommier et al., 2010*). In both screens, we detected a significant enrichment of inactivating insertions in gene *TOP2A*, which encodes topoisomerase IIA (*Figure 1—figure supplement 2a–b*). Bortezomib kills cancer cells by proteasome inhibition (*Adams, 2004*) and, accordingly, we observed a significant enrichment of inactivating insertions in genes encoding proteasome subunits (*Figure 1—figure supplement 2d*). In addition to targets of anticancer drugs, the screens also identified transporters and genes known to metabolize the drug tested (*Figure 1—figure supplement 2*). Thus, loss-of-function screens in KBM7 cells are a powerful way to initiate MOA studies of anticancer compounds.

### Kennedy pathway mutants are resistant to OPA

We then used the KBM7 screening platform to investigate the mechanism of cytotoxicity of OPA, isolating human cells containing insertions that rendered cells resistant to OPA treatment. Three genes had a significant enrichment of retroviral insertions: ethanolamine kinase 1 (*ETNK1*, p = 7.2 × $10^{-12}$), phosphate cytidylyltransferase 2, ethanolamine (*PCYT2*, p = 4.0 × $10^{-7}$), and ethanolamine-phosphotransferase 1 (*EPT1*, p = 4.0 × $10^{-7}$) (*Figure 1a*). These three genes encode the three enzymes required for the de novo synthesis of PE, also known as the Kennedy pathway (*Gibellini and Smith, 2010*) (*Figure 1b*). To test the robustness of this result, we repeated screens at different concentrations of OPA; at least one gene in the Kennedy pathway was enriched above background at every concentration tested (*Figure 1—figure supplement 3*).

We isolated and characterized KBM7 clonal cell lines carrying inactivating insertions in the genes *PCYT2* (in first intron) and *ETNK1* (in first exon) (named PCYT2[GT] and ETNK1[GT], respectively). As expected, these cell lines had strongly reduced levels (<2%) of either *PCYT2* or *ETNK1* mRNA, as quantified by RT-qPCR (*Figure 1c*) and, consistent with the screening results, both clones were less sensitive to OPA (*Figure 1d* and *Figure 1—figure supplement 4a*). Since human cells can synthesize PE through multiple mechanisms (the two major ones being the Kennedy pathway and decarboxylation of phosphatidylserine in mitochondria; *Gibellini and Smith, 2010*), and it is known that cells employ mechanisms to maintain homeostasis of phospholipid levels (*Hermansson et al., 2011*), we tested whether inactivation of the Kennedy pathway in KBM7 cells decreases total cellular PE levels. We extracted total lipids, separated phospholipids by thin layer chromatography, and quantified

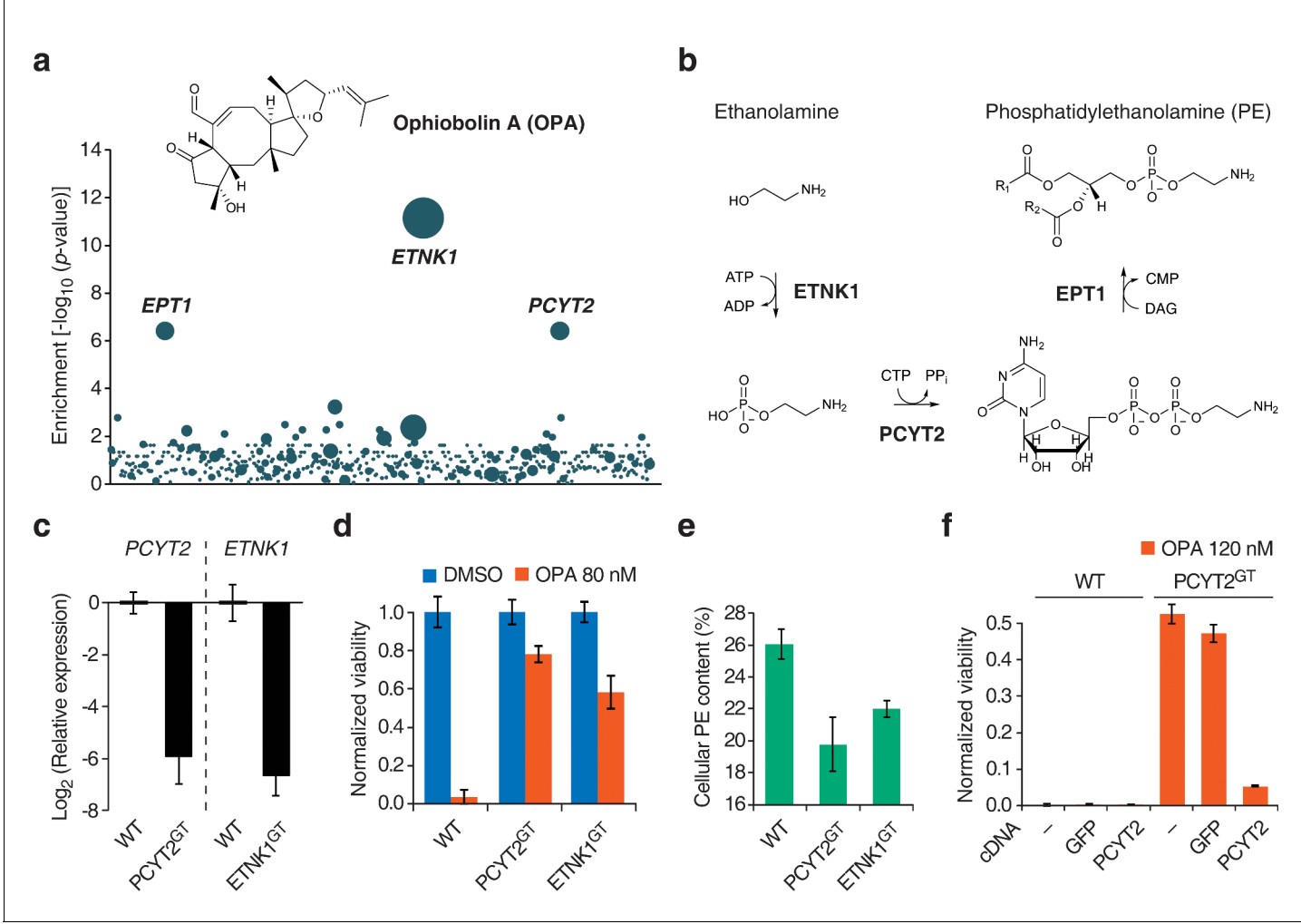

**Figure 1.** Identification of a genetic interaction between ophiobolin A (OPA) and the Kennedy pathway using a loss-of-function genetic screen in the near-haploid human cell line KBM7. (a) A collection of loss-of-function mutants generated in KBM7 cells using retroviral insertional mutagenesis was treated with 388 nM OPA. Resistant clones were allowed to expand for 3 weeks and retroviral insertion sites were identified by high-throughput sequencing. For each gene, an enrichment factor (p-value) was calculated to quantify the enrichment of inactivating insertions in the pool of resistant clones compared to the number existing before selection. Each bubble represents a gene and the diameter of the bubble is proportional to the number of unique insertion sites in the pool of resistant clones (for *ETNK1*, N = 11). Genes are ordered on the x axis by chromosomal location (*Figure 1—source data 1*). (b) The Kennedy pathway: de novo synthesis of phosphatidylethanolamine. (c–e) Characterization of KBM7 clonal cell lines resistant to OPA treatment with inactivating mutations in either *PCYT2* or *ETNK1*, referred to as PCYT2$^{GT}$ and ETNK1$^{GT}$. (c) Quantification of relative *PCYT2* and *ETNK1* mRNA levels by RT-qPCR, normalized to levels in wild-type KBM7 (WT). (d) Cell viability measurement after 72 hr of treatment with OPA (or DMSO vehicle) using a luciferase-based assay quantifying ATP content. The viability of each vehicle-treated cell line was normalized to 1. (e) Determination of cellular phosphatidylethanolamine (PE) content by total lipid extraction, separation of phospholipids by thin layer chromatography and quantification of phospholipids by phosphorus content analysis. PE content is displayed as a percentage of total phospholipids. (f) Expression of PCYT2 in PCYT2$^{GT}$ cells restores OPA sensitivity. Constructs expressing either PCYT2 or GFP (control) were delivered to WT or PCYT2$^{GT}$ cells by lentiviral transduction. The viability of each cell line was assayed using a luciferase-based assay quantifying ATP content after 72 hr of treatment with OPA (or DMSO vehicle). '—' denotes non-transduced cell lines. (c–f) Results were obtained from three independent experiments (c and e) or from assays performed in triplicate (d and f) and data represent mean values ± standard deviation.

The following source data and figure supplements are available for figure 1:

**Source data 1.** Source data for the ophiobolin A (OPA) loss-of-function KBM7 screen.

**Figure supplement 1.** Illustration of loss-of-function genetic screens in haploid human KBM7 cells (*Carette et al., 2009, 2011*).

**Figure supplement 1—source data 1.** Source data for the characterization of mutagenized KBM7 cells before selection (control library).

*Figure 1 continued on next page*

*Figure 1 continued*

**Figure supplement 2.** Validation of KBM7 loss-of-function screens using anticancer drugs with well-characterized mechanisms of action.

**Figure supplement 2—source data 1.** Source data for anticancer drug loss-of-function KBM7 screens.

**Figure supplement 3.** Loss-of-function genetic screens in KBM7 cells performed at three different concentrations of ophiobolin A (OPA) consistently identify genes in the Kennedy pathway.

**Figure supplement 3—source data 1.** Source data for additional ophiobolin A (OPA) loss-of-function KBM7 screens.

**Figure supplement 4.** Titration of the toxicity of ophiobolin A (OPA) towards KBM7 wild-type and gene-trapped cell lines, and additional data for complementation assays.

phospholipids by phosphorus content analysis. Both PCYT2$^{GT}$ and ETNK1$^{GT}$ cell lines showed reduced PE levels compared to KBM7 cells, by 24% and 16%, respectively (*Figure 1e*). Since gene-trapping of *PCYT2* renders KBM7 cells slightly more resistant to OPA and reduces PE levels to a larger extent than inactivation of *ETNK1*, our further studies focused on *PCYT2*.

To validate that inactivation of the Kennedy pathway causes OPA resistance, we transduced PCYT2$^{GT}$ cells with a lentiviral construct driving expression of PCYT2 and observed that this complementation rescues OPA sensitivity (*Figure 1f*). As expected, both *PCYT2* mRNA levels and total PE content were also restored to wild-type levels in complemented cells (*Figure 1—figure supplement 4b–c*). We next assessed the generality of our results across cell lines by testing the effect of silencing *PCYT2* in HEK293T cells on viability during OPA treatment. Cells expressing two shRNA (short hairpin RNA) constructs had 85% and 70% reduced *PCYT2* mRNA levels, respectively (*Figure 2a*), exhibited increased OPA resistance (*Figure 2b*), and had reduced cellular PE levels (*Figure 2c*). In summary, these data show that a reduction of the Kennedy pathway activity, and thus PE levels, in human cells leads to an increase in OPA resistance.

## OPA treatment results in activation of the Kennedy pathway in human cells

Having established a clear link between the activity of the Kennedy pathway and the cytotoxicity of OPA, we next investigated the underlying molecular mechanism. There is precedent for small molecules that target the Kennedy pathway: the antihistamine and antiemetic drug meclizine directly inhibits PCYT2, reducing the average flux through the Kennedy pathway (*Gohil et al., 2013*). To explore the possibility that OPA exerts cytotoxicity by inhibiting or activating one of the enzymes in the Kennedy pathway, we measured the activity of the pathway after OPA treatment in the two commonly used human cell lines, HEK293T and HCT116. Cells were treated with OPA (or vehicle) and then incubated with ethanolamine [1,2-$^{14}$C], the substrate of the first enzyme in the Kennedy pathway. Interestingly, when either cell line was treated with concentrations of OPA that induced mild cytotoxicity, both the average flux through the Kennedy pathway (measured by accumulation of the radiolabel into PE) and the steady state level of PE were *increased* compared to vehicle-treated cells (*Figure 2d*). In contrast, when we treated cells with meclizine we observed the expected decrease in the flux through the pathway and reduced PE content (*Figure 2d*). Since the change in enzyme activity induced by OPA treatment is small, we do not believe that OPA acts directly on an enzyme in the Kennedy pathway to cause cytotoxicity.

## OPA is inactivated by exogenous PE in cell culture medium

The known reactivity of OPA with primary amines (*Au et al., 2000*) and the observation that resistance to OPA toxicity correlates with lower PE content (*Figures 1e* and *2c*) led us to propose that OPA might directly target PE through covalent modification of its ethanolamine head group. If this hypothesis is correct, adding exogenous PE along with OPA to the growth medium of cells should result in the formation of PE-OPA covalent adducts in the medium. The majority of these adducts should not partition

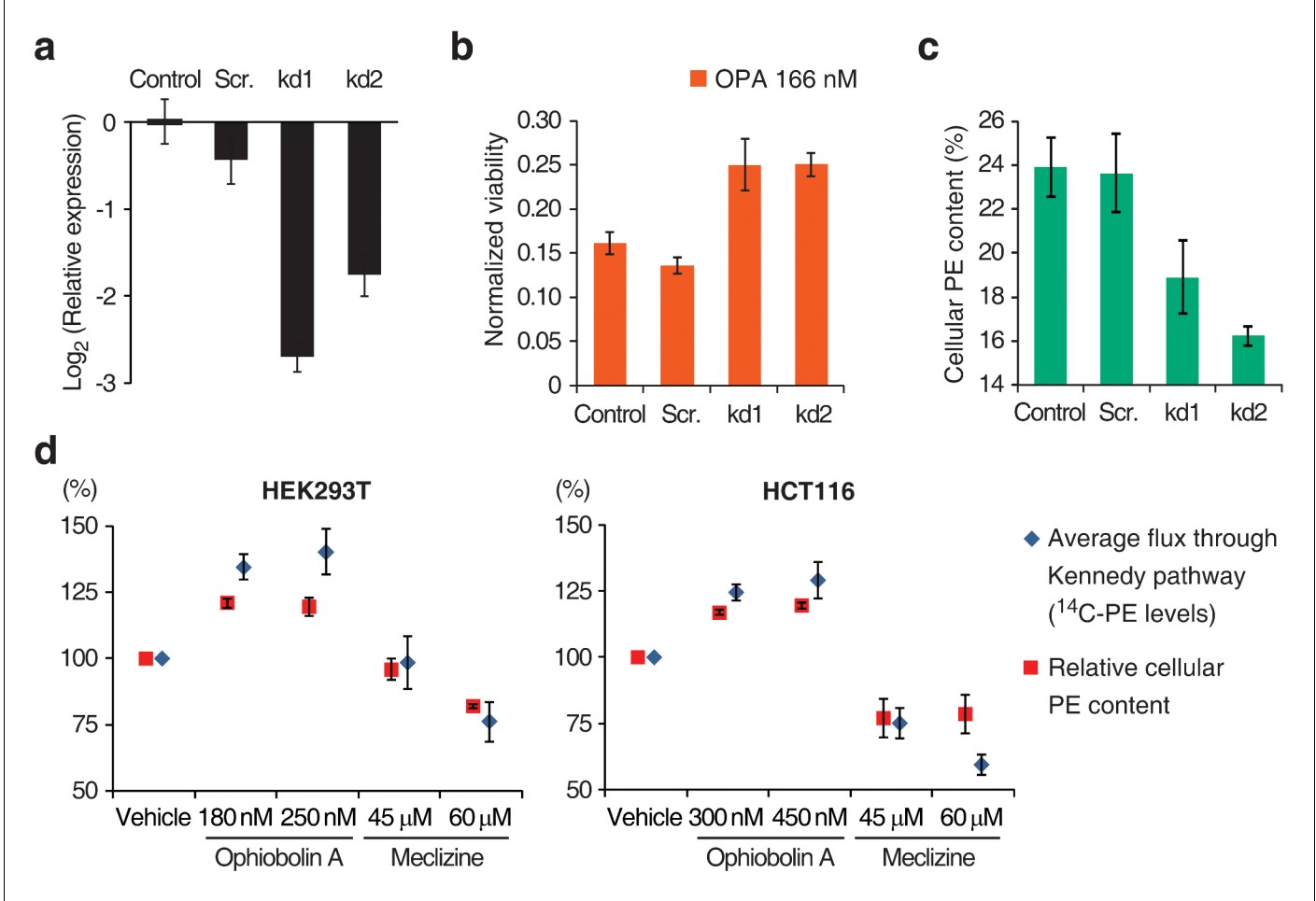

**Figure 2.** Interaction between the activity of the Kennedy pathway and ophiobolin A (OPA) cytotoxicity. (**a–c**) shRNA knockdown of the Kennedy pathway in HEK293T cells leads to increased resistance to OPA toxicity. Constructs enabling stable expression of shRNAs against *PCYT2* (kd1 and kd2), a scrambled shRNA (Scr.), or an empty vector control (Control) were delivered to HEK293T cells by lentiviral transduction. Results were obtained from assays performed on three independent transduced cell lines and data represent mean values ± standard deviation. (**a**) Quantification of relative *PCYT2* mRNA levels by RT-qPCR, normalized to the level of *PCYT2* mRNA in the control. (**b**) Cell viability measurement after 72 hr of treatment with OPA (or DMSO vehicle) using a luciferase-based assay quantifying ATP content. The viability of each vehicle-treated cell line was normalized to 1. (**c**) Determination of cellular phosphatidylethanolamine (PE) levels by total lipid extraction, separation of phospholipids by thin layer chromatography (TLC), and quantification of phospholipids by phosphorus content analysis. PE content is displayed as a percentage of total phospholipids. (**d**) OPA treatment activates the Kennedy pathway and increases PE content in HEK293T and HCT116 cells. Cells were treated with OPA, meclizine, or DMSO vehicle for 5 hr, then ethanolamine [1,2-$^{14}$C] was added and the treatment was prolonged for an additional 24 hr. Total phospholipids were extracted and separated by silica gel TLC. PE contents were quantified as in (**c**) and the PE content of vehicle-treated cells was normalized to 100%. $^{14}$C-PE levels were quantified by liquid scintillation counting of silica scrapings and were normalized to total phospholipid content. Results were obtained from three independent experiments and data represent mean values ± standard error of the mean.

efficiently into cells due to limited aqueous solubility, and thereby adding exogenous PE should reduce the number of OPA molecules available to react with endogenous PE and kill cells. In agreement with this prediction, addition of a commercial preparation of PE to cells treated with a cytotoxic amount of OPA rescued cell viability, whereas addition of the phospholipids phosphatidylcholine (PC) or phosphatidylserine (PS) had no effect (*Figure 3a*). PE extracts from diverse sources were all similarly able to rescue cellular viability, suggesting that small quantities of impurities in these preparations are not likely responsible for OPA inactivation (*Figure 3—figure supplement 1a*). To provide additional evidence that the difference in the effects of PE and PC on OPA toxicity derive solely from differences in the head group (and not differences in fatty acid composition or impurities), we assayed a pair of

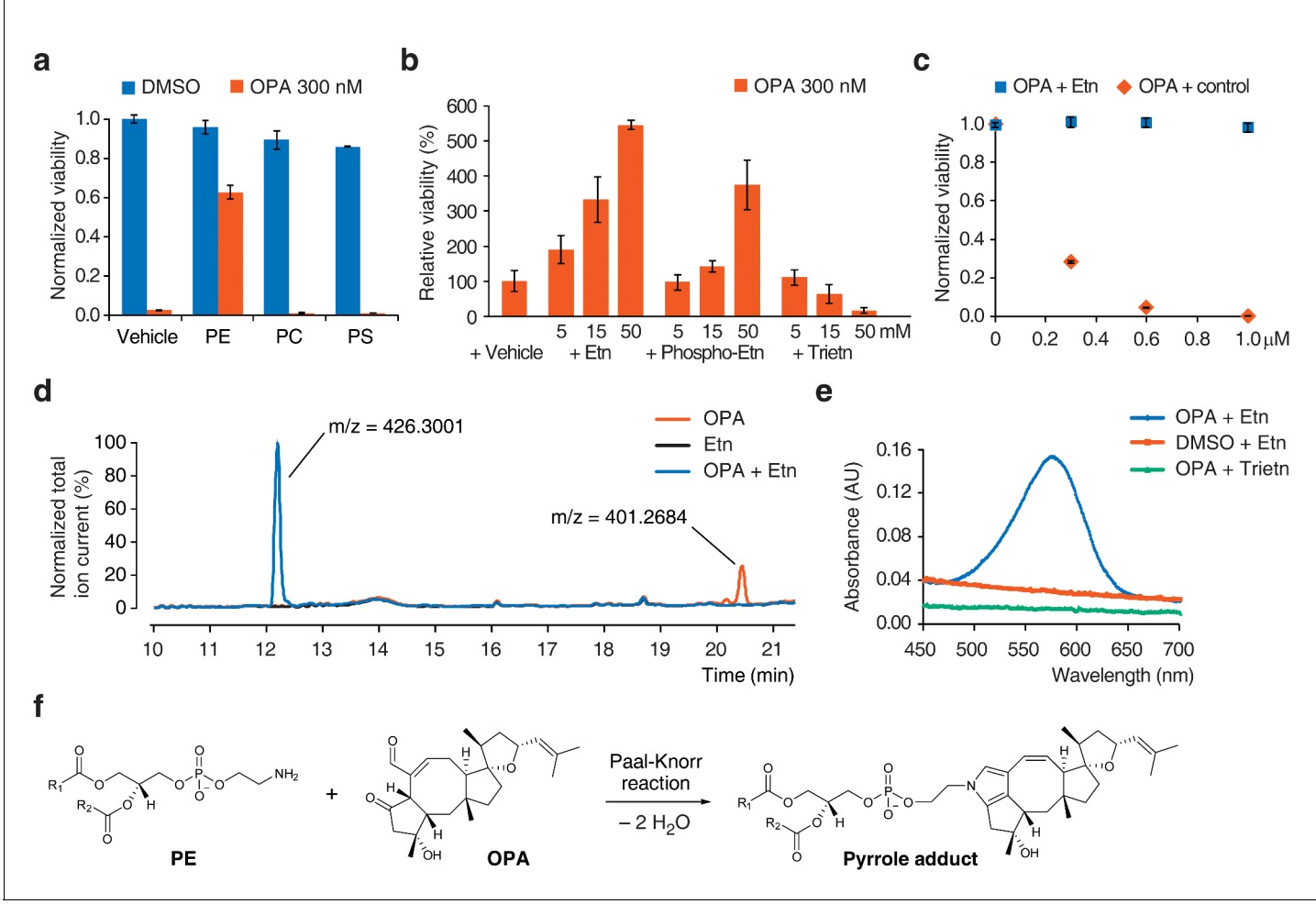

**Figure 3.** Ophiobolin A (OPA) reacts with the ethanolamine (Etn) head group of phosphatidylethanolamine (PE) via a Paal-Knorr reaction. (**a**) Exogenous PE, but not phosphatidylcholine (PC) or phosphatidylserine (PS), added to growth medium quenches the cytotoxicity of OPA. Commercially available phospholipids extracted from chicken egg (PE and PC) or bovine brain (PS) (or vehicle) were added to 20 ng/µL (~25 µM) to the growth medium of HEK293T cells. Cells were subsequently treated with OPA (or DMSO vehicle) for 72 hr and cell viability was then quantified using a luciferase-based assay measuring ATP content. The viability of vehicle-treated cells in the absence of OPA was normalized to 1. (**b**) The primary amine of Etn is essential for OPA inactivation. A cell viability assay was performed as in (**a**) with Etn, *O*-phosphorylethanolamine (Phospho-Etn) and triethanolamine (Trietn). Only viabilities in 300 nM OPA are displayed and the viability of vehicle-treated cells in 300 nM OPA was normalized to 100%. Full plots are available in *Figure 3—figure supplement 1b*. (**c**) OPA was incubated with an excess of Etn (or ethanol control) in aqueous buffer. HEK293T cells grown in standard conditions were treated with the reaction product for 72 hr and cell viability was then quantified using a luciferase-based assay measuring ATP content. (**a–c**) Results were obtained from assays performed in triplicate and data represent mean values ± standard deviation. (**d**) Liquid chromatography-mass spectrometry (LC-MS) analysis in positive ion mode of the in vitro reaction of OPA (exact mass = 400.2614) with Etn (exact mass = 61.0528) shows formation of a single product at an m/z corresponding to an addition reaction minus two molecules of $H_2O$. Both OPA and Etn were used as reactant controls and the total ion chromatograms of the three samples are displayed overlaid. The m/z of the most abundant ion is displayed above corresponding peaks. (**e**) OPA reacts with Etn to form a pyrrole-containing product detectable using Ehrlich's reagent. An in vitro reaction of OPA with Etn was mixed with Ehrlich's reagent and the absorbance of the resulting solution was measured between 450 and 700 nm. Reactions with DMSO (vehicle) instead of OPA and Trietn instead of Etn were used as negative controls. (**f**) Proposed reaction between PE and OPA.

The following figure supplements are available for figure 3:

**Figure supplement 1.** Additional data for ophiobolin A (OPA) inactivation assays with exogenously added small molecules.

**Figure supplement 2.** Proposed mechanism of covalent modification of phosphatidylethanolamine (PE) by ophiobolin A (OPA) through a Paal-Knorr-like reaction pathway (*Bernoud-Hubac et al., 2004*; *Amarnath et al., 1991*, *1995*).

commercial lipid preparations: one that consists of an extract of PC, and an identical PC extract in which the choline head group of PC has been exchanged for ethanolamine to yield a transphosphatidylated PE extract. When tested in the OPA inactivation assays, transphosphatidylated PE rescued cell viability whereas PC did not, substantiating the claim that PE is the molecule responsible for OPA inactivation (*Figure 3—figure supplement 1a*). In addition, we observed that constituents of the head group of PE, ethanolamine and *O*-phosphorylethanolamine, could inactivate OPA, although much less potently than PE, and that triethanolamine (lacking a primary amine) had no effect (*Figure 3b* and *Figure 3—figure supplement 1b*).

The serine head group of PS also contains a primary amine that could react with OPA. However, we observed no inactivation of OPA using either a natural extract of PS (*Figure 3a*) or synthetic dioleoyl-PS (*Figure 3—figure supplement 1c*). In contrast to ethanolamine, adding serine to the growth medium of cells does not lead to OPA inactivation, suggesting that the primary amine in the serine head group is less reactive with OPA than that in the ethanolamine head group (*Figure 3—figure supplement 1d*). This observation is consistent with previous reports of inefficient adduct formation between PS and reactive aldehydes from fatty acid peroxidation (*Guichardant et al., 1998*, *2002*). In conclusion, these results suggest that OPA is specifically inactivated by PE via reaction with the primary amine on its head group.

## OPA reacts with ethanolamine in vitro and forms a pyrrole-containing covalent adduct

Synthetic studies exploring the chemical reactivity of OPA have shown that OPA can react with primary amines in a Paal-Knorr reaction to yield pyrrole-containing adducts (*Dasari et al., 2015*). To test the hypothesis that OPA also reacts with PE through its primary amine, we explored the product of the reaction between OPA and ethanolamine. Incubation of OPA with an excess of ethanolamine abolished cytotoxicity and analysis by LC-MS/MS revealed that the main product of this reaction is consistent with formation of a covalent adduct with two dehydration reactions (*Figure 3c–d*). Incubation of the reaction product of OPA and ethanolamine with Ehrlich's reagent (*Amarnath et al., 2004*) yielded a purple solution ($A_{max}$ = 580 nm), characteristic of pyrroles (*Figure 3e*). Addition of salicylamine and pentyl-pyridoxamine, potent scavengers of 1,4-dicarbonyls, to OPA led to its inactivation at concentrations 30- and 300-fold lower than for ethanolamine, consistent with the known kinetics of these scavengers in Paal-Knorr reactions (*Amarnath et al., 2004*, *2015*) (*Figure 3—figure supplement 1e*). These findings, together with the results from the OPA inactivation assays, are consistent with OPA reacting with PE according to a Paal-Knorr reaction mechanism (*Figure 3f* and *Figure 3—figure supplement 2*).

## OPA reacts with PE in vitro and in human cells

Based on our observations that OPA can be inactivated by exogenous PE in vitro and that it forms a pyrrole-containing adduct with ethanolamine, we hypothesize that the bioactivity of OPA arises from the formation of covalent adducts with PE. To facilitate detection of such adducts in human cells after treatment with OPA, we utilized phospholipase D (PLD) from *Streptomyces chromofuscus* (*Sullivan et al., 2010*) to release modified PE head groups from heterogeneous populations of lipids differing in fatty acid composition (*Figure 4a*). We first confirmed the efficacy of this approach by synthesizing PE-OPA covalent adducts in vitro and hydrolyzing these adducts with PLD to release ethanolamine-OPA (Etn-OPA) (*Figure 4b*). Etn-OPA was unequivocally characterized by its exact mass, retention time, and MS/MS fragmentation pattern using the reaction product of OPA with ethanolamine as a standard (*Figure 3d*). The detection of Etn-OPA was dependent on the presence of PE and OPA, and PLD treatment, demonstrating that OPA forms a pyrrole-containing covalent adduct with PE which PLD is able to hydrolyze into Etn-OPA (*Figure 4b*).

We next used this analytical strategy to query the presence of PE-OPA adducts in the pool of phospholipids extracted from human cells treated with OPA. HEK293T and HCT116 cells were treated, total cellular phospholipids were extracted, and residual OPA was rapidly quenched with pentyl-pyridoxamine to prevent any reaction of PE with residual OPA after cell lysis (*Amarnath et al., 2015*). The quenched extracted phospholipids were digested with PLD and analyzed by LC-MS/MS, which revealed the presence of Etn-OPA dependent on OPA and PLD treatment (*Figure 4c–d* and *Figure 4—figure supplements 1* and *2*). Importantly, the absence of Etn-

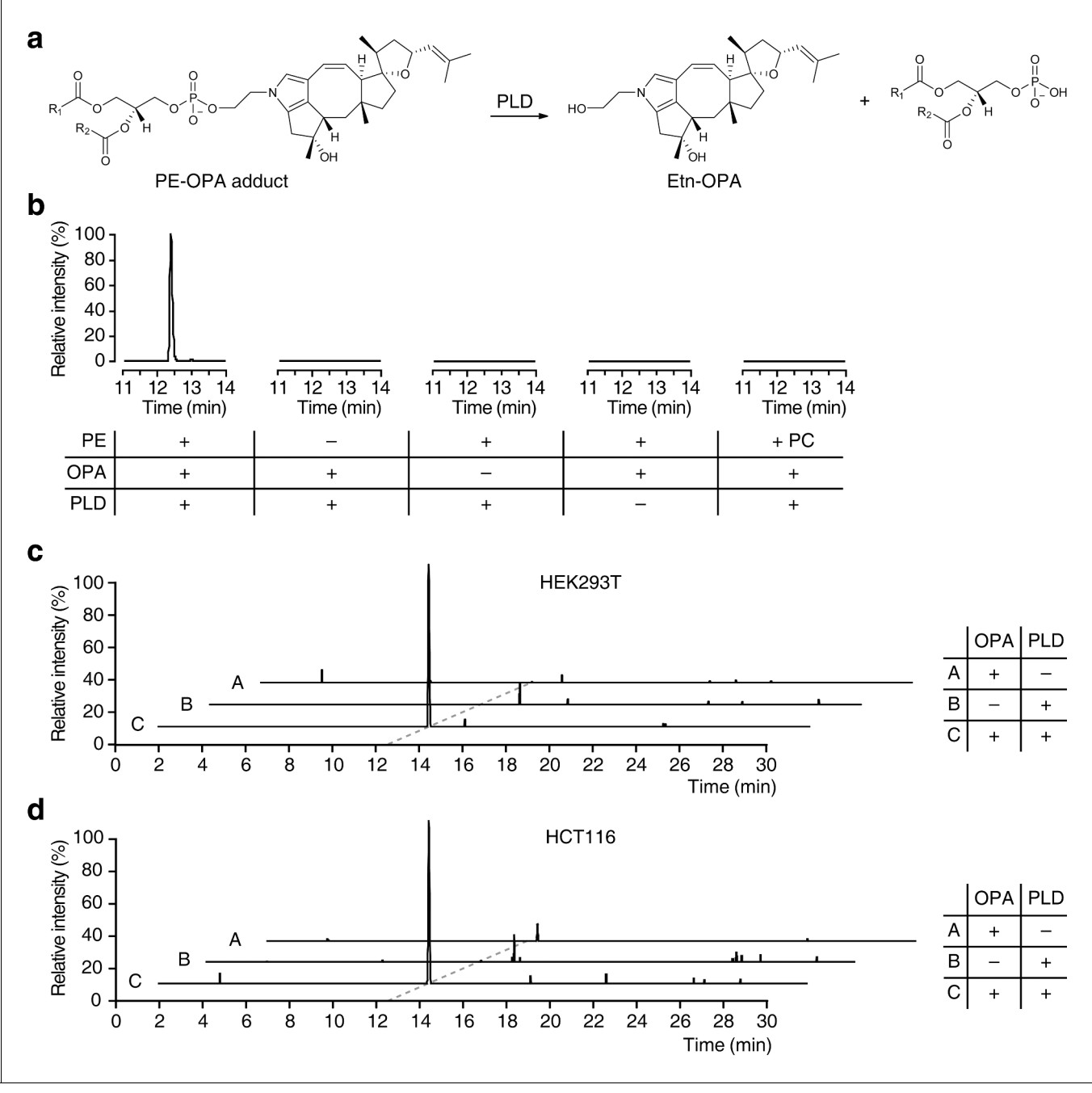

**Figure 4.** Ophiobolin A (OPA) forms a pyrrole-containing covalent adduct with phosphatidylethanolamine (PE) in human cells. (a) Formation of PE-OPA adducts was detected by measuring the abundance of ethanolamine-OPA (Etn-OPA) after hydrolysis by phospholipase D from *Streptomyces chromofuscus* (PLD). (b) Extracted ion chromatograms (m/z = 426.2982–426.3024) of the liquid chromatography-mass spectrometry (LC-MS) analysis of in vitro reactions of PE with OPA and subsequent digestion with PLD. Control reactions include systematic replacement of each reagent by vehicle and replacement of PE with phosphatidylcholine (PC). (c–d) Extracted ion chromatograms (m/z = 426.2982–426.3024) showing the detection of PE-OPA adducts in lipids extracted from cells treated with OPA. (c) HEK293T cells grown in standard conditions were incubated with 250 nM OPA for 24 hr. Total cellular lipids were extracted in the presence of pentyl-pyridoxamine to quench unreacted OPA. Lipids were incubated with PLD and analyzed by LC-MS for the presence of Etn-OPA. Negative controls include replacement of OPA by DMSO vehicle or absence of PLD treatment. (d) Same as (c) but for HCT116 cells treated with 450 nM OPA. Full chromatograms and replicate experiments are available in *Figure 4—figure supplement 1*.

The following figure supplements are available for figure 4:

**Figure supplement 1.** Raw data and replicate experiment for data represented in *Figure 4c–d*.

*Figure 4 continued on next page*

*Figure 4 continued*

**Figure supplement 2.** Higher-energy collisional dissociation (HCD) MS/MS fragmentation spectra of ethanolamine-ophiobolin A (Etn-OPA).

OPA in the control lacking PLD treatment indicates that OPA does not react with cellular ethanolamine (*Figure 4c–d*). These findings demonstrate that OPA forms pyrrole-containing covalent adducts with PE in human cells.

## OPA induces membrane leakiness

Modification of PE to PE-OPA adducts substantially changes the biophysical properties of PE by modifying its head group from small and polar to bulky and hydrophobic. We thus hypothesized that PE-OPA adduct formation could lead to destabilization of cellular lipid bilayers and be the main cause of OPA cytotoxicity. To determine whether OPA induces membrane leakiness due to adduct formation, we used artificial liposomes whose PE content we could modulate. We prepared large unilamellar lipid vesicles (LUVs) composed of a variable ratio of dioleoyl-PE to dioleoyl-PC and loaded with the fluorescent probe calcein (*Allen and Cleland, 1980*; *Zhang et al., 2001*). As calcein fluorescence becomes dequenched upon release from LUVs, an increase in fluorescence can be used as an indicator of liposome leakage. We observed that OPA treatment caused extensive liposome leakage, and that the extent of leakage is directly dependent on both the PE content of the liposomes and on the OPA concentration (*Figure 5a*). Importantly, we observed that liposomes composed only of PC or with very low PE content remain intact even at high OPA concentration, indicating that the leakage is likely specifically due to the formation of PE-OPA adducts. These results clearly show that OPA destabilizes lipid bilayers, supporting our hypothesis that membrane permeabilization is the main cause of OPA cytotoxicity.

## Discussion

Our work illustrates the utility of loss-of-function screens in human cells to identify genes involved in the MOA of promising anticancer compounds such as OPA. The Kennedy pathway genes identified in the screen of OPA led to the identification of a phospholipid molecule as the cellular target of OPA. This discovery was possible due to the unbiased nature of the screen and would have been difficult to achieve using alternative methods such as affinity chromatography followed by mass spectrometry. Few studies have yet taken advantage of the recent development of novel genetic tools in human cells to identify the target of small molecules, and this is the first study to uncover a non-protein based target (*Nijman, 2015*). Here we show that OPA forms a pyrrole-containing covalent adduct with PE in human cells. Reduction of PE levels through inactivation of the Kennedy pathway results in a reduction in OPA cytotoxicity. We also show that PE-OPA adduct formation leads to destabilization of lipid bilayers in vitro. Collectively, this study indicates that PE is the main target of OPA in human cells and leads to the hypothesis that formation of PE-OPA adducts directly causes the observed cytotoxicity of OPA through membrane destabilization (*Figure 5b*).

OPA is reactive towards primary amines, but the chemical reactivity of PE alone does not explain why OPA would selectively react with PE over ethanolamine, lysine side chains of proteins, or any other abundant primary amine in human cells (*Dasari et al., 2015*). While we cannot exclude the possibility that covalent modification of cellular proteins contributes to the observed toxicity of OPA, our experimental results show that the concentration of PE required to inactivate OPA in vitro is at least 2000-fold lower than that of ethanolamine or lysine (*Figure 3a* and *Figure 3—figure supplement 1*). This selectivity may arise because, at the concentrations tested, ethanolamine and lysine are freely soluble in aqueous buffers whereas PE forms insoluble lipid aggregates. Considering the lipophilic nature of OPA, it is likely that OPA accumulates in the lipid aggregates and efficiently reacts with PE due to high local concentrations or to the hydrophobic environment, or both. In a comparable way, we believe that the selectivity in living cells is the result of accumulation of OPA in lipid bilayers and efficient reaction with PE, the most abundant primary amine in human lipid bilayers (*Vance and Tasseva, 2013*). The observation that OPA does not react with ethanolamine in human cells, despite efficient reaction with PE, supports this claim (*Figure 4c–d*).

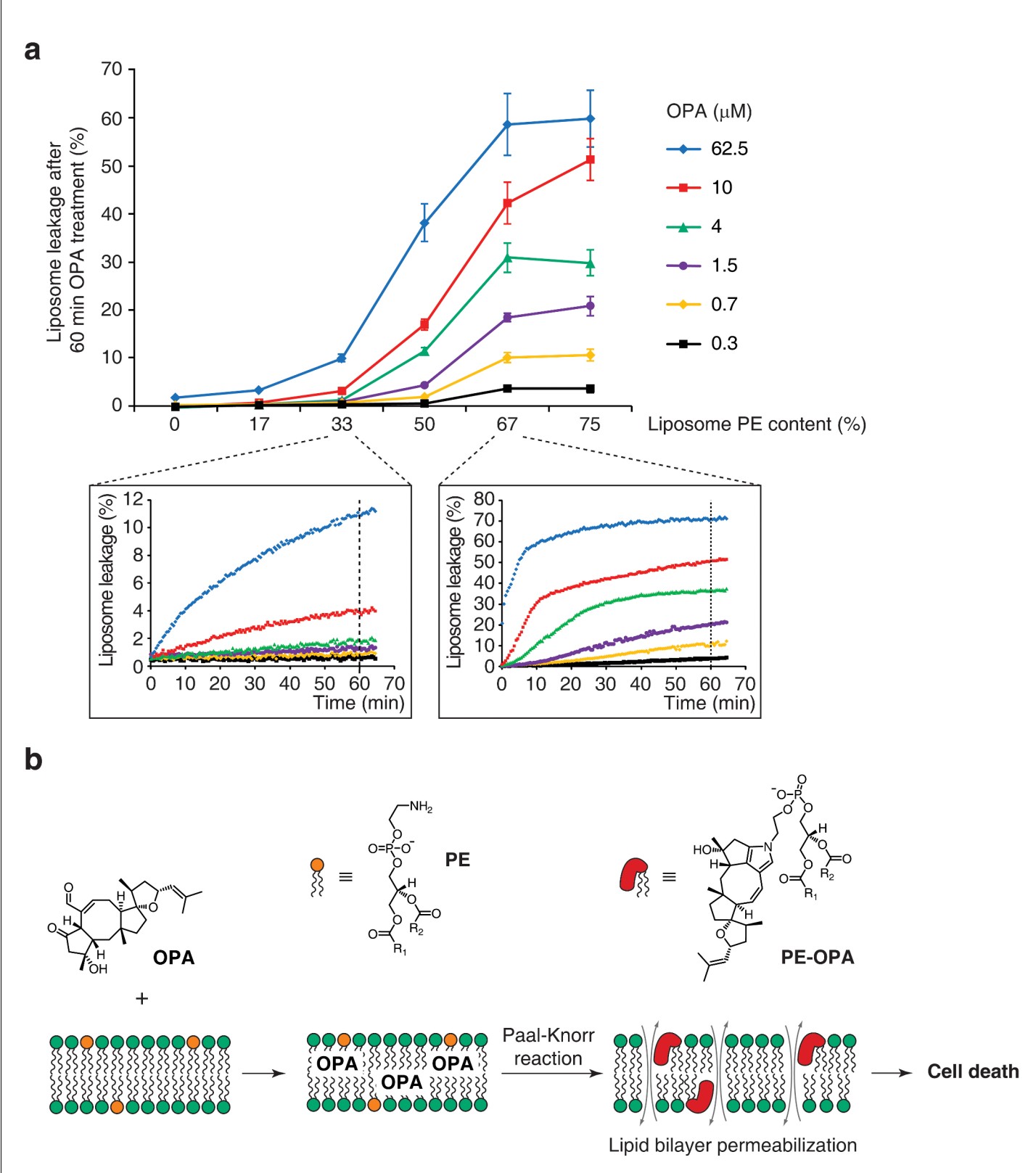

**Figure 5.** Ophiobolin A (OPA) induces leakage from liposomes. (**a**) Effect of 60 min OPA treatment on the leakage of the fluorescent dye calcein from artificial liposomes composed of various ratios of phosphatidylethanolamine (PE) to phosphatidylcholine (PC). Large unilamellar vesicles (LUVs) composed of dioleoyl-PE and dioleoyl-PC, and encapsulating calcein were prepared by extrusion. Leakage assays were initiated by addition of OPA

*Figure 5 continued on next page*

*Figure 5 continued*

and were monitored by quantification of the fluorescence of calcein, which dequenches upon release from liposomes. OPA-induced liposome leakage was determined by normalization of fluorescence to a DMSO vehicle control (0% leakage) and to detergent-treated liposomes (100% leakage). Results were obtained from assays performed in triplicate and data represent mean values ± standard error of the mean. Inset plots display examples of the kinetics of leakage after OPA addition for two liposome preparations. (**b**) Proposed model of the mechanism of action of OPA in human cells. When applied to cells, OPA accumulates in the phospholipid bilayer of the plasma membrane as it is a lipophilic compound. Due to high local concentrations or the hydrophobic environment (or both), OPA efficiently reacts with the primary amine head group of phosphatidylethanolamine (PE) in a Paal-Knorr-like reaction. Formation of PE-OPA adducts changes the biophysical properties of PE by modifying its head group from small and polar to bulky and hydrophobic, leading to membrane permeabilization and ultimately cell death.

The increase in OPA resistance conferred by inactivation of the Kennedy pathway in our haploid genetic screen strongly suggests that PE-OPA covalent adduct formation is a major determinant of OPA cytotoxicity. We surmise that the observed cytotoxicity of OPA reflects the amount of covalent PE-OPA adducts formed in cells, which, based on the mechanism of the Paal-Knorr reaction, should be dependent on the concentrations of both OPA and PE. Accordingly, we observed that a mild reduction in PE levels (16–24%) was accompanied by a mild increase in OPA resistance (IC$_{50}$ from 43 nM in wild-type cells to 70–85 nM in the mutants) (*Figure 1e* and *Figure 1—figure supplement 4a*). As the amplitude of changes in the observed cytotoxicity of OPA can be accounted for by changes in PE levels, we believe that PE represents the main target of OPA in human cells.

Several members of the ophiobolin family of fungal metabolites have been isolated and evaluated for cytotoxicity in a panel of human cell lines (*Au et al., 2000*; *Dasari et al., 2015*). Notably, the C-6 epimer of OPA (6-*epi*-OPA) is 40-fold less cytotoxic than OPA (*Dasari et al., 2015*), although both compounds would form the same pyrrole adduct upon reaction with PE, as the stereocenter at position 6 is lost with pyrrole formation. This observation must mean that the reaction of PE with 6-*epi*-OPA is on the order of 40-fold slower than the reaction of PE with OPA. The ring cyclization step is the rate limiting step of the Paal-Knorr reaction (*Figure 3—figure supplement 2*) and this step is strongly influenced by the stereochemistry of the 1,4 diketone; rates of pyrrole formation can be up to 57-fold different between stereoisomers (*Amarnath et al., 1991*). Thus, the 40-fold difference in potency between OPA and 6-*epi*-OPA can likely be attributed to differences in the rate of pyrrole adduct formation, and this observation supports our model that cytotoxicity depends on the amount of PE-OPA adduct formed.

We hypothesize that formation of PE-OPA adducts disrupts the lipid bilayer of human cells to induce cell death. As seen in the LUVs leakage experiments, OPA induces strong membrane permeabilization in model membranes and the extent of permeabilization is dependent on PE content (*Figure 5a*). Interestingly, this observation is consistent with our expectation that the rate of adduct formation is dependent on PE content and suggests that OPA might display higher potency against cells or tissues containing high PE contents. Also, the dependence of membrane leakiness on PE content provides a mechanistic basis for our observation that Kennedy pathway KBM7 mutants are resistant to OPA due to lower PE levels. Cytotoxicity through modification of PE has previously been reported for polygodial, a 1,4-dialdehyde antifungal compound, which has been shown to form a pyrrole-containing adduct with PE on the cell surface that is directly linked to its antifungal activity (*Fujita and Kubo, 2005*). Furthermore, it has been previously shown that covalent modification of PE with isoketals alters membrane curvature because the newly formed bulky hydrophobic head group partitions to the lipid bilayer and increases lateral pressure (*Guo et al., 2011*). We believe such a mechanism may contribute to the ability of PE-OPA adducts to disrupt lipid bilayers, and suspect that the extent of lipid bilayer destabilization is highest in the plasma membrane as it is the first source of PE encountered by OPA.

We were initially surprised to find that treatment of human cells with OPA led to activation of the Kennedy pathway and an increase in PE content (*Figure 2d*). However, phospholipid homeostasis in mammalian cells is not well understood, but it is generally accepted that mechanisms are in place to tightly regulate membrane lipid composition (*Hermansson et al., 2011*). The activation of the pathway seen upon OPA treatment may be a cellular response to the formation of PE-OPA adducts due to mechanisms regulating membrane PE homeostasis.

In addition to its activity in human cells, OPA is cytotoxic towards a broad range of organisms (*Au et al., 2000*). Early studies on the mechanism of cytotoxicity of OPA in plant cells suggested that it causes non-specific damage to membranes (*Chattopadhyay and Samaddar, 1976*) or even "covalent modification of some membrane component" (*Tipton et al., 1977*). Considering our work in human cells and the fact that PE is ubiquitously found in nature (*Vance and Tasseva, 2013*), PE may be the main target of OPA in plants and potentially other organisms.

OPA has been shown to have an antitumor effect in a mouse glioblastoma model (*Bury et al., 2013*; *Dasari et al., 2015*). The basis for the tumor selectivity of OPA may be due to altered distribution or higher abundance of PE in cancer cells. Indeed, it has recently been shown that PE is found in higher quantities on the outer leaflet of cancer cells (*Stafford and Thorpe, 2011*). Furthermore, host defense peptides have been shown to display selectively against cancer cells based on the difference in surface phospholipid composition compared to normal cells (*Leite et al., 2015*; *Riedl et al., 2011*). Using the changes in lipid composition of cancer cells as a biomarker represents an interesting approach to chemotherapeutics development (*Leite et al., 2015*) and our findings raise the exciting possibility that OPA will prove an effective chemotherapy tool for multidrug-resistant glioblastoma.

## Materials and methods

### Chemical reagents

Unless otherwise mentioned, all chemicals were from Sigma-Aldrich (St Louis, MO). Ophiobolin A (>95%) was from Enzo Life Sciences (Farmingdale, NY). Gemcitabine, oxaliplatin, bortezomib, topotecan, and doxorubicin were from LC Laboratories (Woburn, MA). Ethanolamine [1,2-$^{14}$C] HCl was from American Radiolabeled Chemicals (St Louis, MO). Phospholipids were from Sigma-Aldrich (PE, chicken egg; PC, chicken egg; PS, bovine brain) and from Avanti Polar Lipids (Alabaster, AL) (transphosphatidylated PE, chicken egg; PE, porcine brain; DOPE; DOPC; DOPS). Salicylamine and pentyl-pyridoxamine were a kind gift of V. Amarnath (Vanderbilt) (*Amarnath et al., 2004*, *2015*). Stock solutions, unless otherwise mentioned, were prepared in DMSO (99.9%) at 500× concentration to yield a final concentration of 0.2% DMSO.

### Cell culture

KBM7 cells were obtained from Thijn Brummelkamp (*Carette et al., 2009*, *2011*) and were cultured at 37°C in 5% $CO_2$ in Iscove's Modified Dulbecco Medium (IMDM) (Gibco, Thermo Fisher Scientific, Waltham, MA) supplemented with 10% heat inactivated fetal bovine serum (Gibco), and penicillin/streptomycin at final concentrations of 100 U/mL and 100 μg/mL, respectively (P/S) (Corning Inc., Corning, NY). HEK293T and HCT116 were obtained from ATCC (Manassas, VA). HEK293T were cultured at 37°C in 5% $CO_2$ in Dulbecco's Modified Eagle Medium (DMEM) (ATCC) supplemented with 10% fetal bovine serum (FBS) (ATCC), and P/S. HCT116 were cultured at 37°C in 5% $CO_2$ in McCoy's 5A (ATCC) supplemented with 10% FBS, and P/S. Cell lines were used at low passage numbers from primary stocks and were not further authenticated or tested for mycoplasma.

### Haploid genetic screens in KBM7 cells

Mutagenized KBM7 cells were prepared as in *Birsoy et al. (2013)*. For each screen, 100 million cells were diluted in 200 mL growth medium and the small molecule of interest was added from a 500× stock solution in DMSO. Then 100,000 cells per well were aliquoted in 96-well plates. Plates were incubated at 37°C in 5% $CO_2$ until colonies were visible (about 3 weeks). All resistant cells were pooled, washed with Dulbecco's phosphate buffer saline (PBS) and genomic DNA was prepared from 30 million cells using the QIAamp DNA mini kit (Qiagen, Hilden, Germany). Genomic DNA was first digested in separate reactions with NlaIII and MseI and then self-ligated under dilute conditions using T4 DNA ligase (New England Biolabs, Ipswich, MA). After clean-up of the reactions using the MiniElute PCR purification kit (Qiagen), self-ligated products were amplified in a PCR reaction using 10 μM LTRSolexaI, 1 μM either NlaIII or MseI adaptor (depending on the enzyme used for DNA digestion), and 10 μM of index primer, and using Phusion Hot Start Flex polymerase (New England Biolabs). PCR products were cleaned up using the MiniElute PCR purification kit and the presence of amplified products was verified by agarose gel electrophoresis. The PCR reactions of up to 20 screens with unique barcodes were pooled and sequenced using Illumina's (San Diego, CA) HiSeq

2500 platform (50 bp, single read) and using primer SolexaSeqFlank. About 5–10 million reads were obtained for each screen.

Reads containing MseI or NlaIII sites flanked by vector DNA sequence were trimmed after the restriction site (to allow potential alignment of fragments shorter than 50 bp). Using Bowtie (*Langmead et al., 2009*), reads were aligned to the human genome hg19 with no mismatch allowed and a single alignment site. A list of unique genomic alignment sites was compiled and sites separated by only 1 or 2 bp were discarded. Additionally, alignment sites represented by only one sequencing read were discarded. The insertion sites were next compared to a list of all annotated human introns and exons (Roche Nimblegen Exon-Intron table, July 2010, hg19). For each human gene, the total number of unique insertions in exonic regions and those in intronic regions in the sense orientation were counted. Finally, an enrichment p-value was calculated using Fisher's exact test for each annotated gene by comparing the number of inactivating insertion sites after selection to the number of inactivating insertions in that gene in a control library. Genes with less than 10 total reads were not displayed in the bubble plots.

A control library was prepared by extraction of genomic DNA from mutagenized KBM7 cells collected prior to initiating loss-of-function screens. PCR products were prepared and analyzed in the same way as above except that genomic DNA from a total of 5 million cells was used as template and 35 million sequencing reads were obtained.

## Identification and preparation of clonal gene-trapped KBM7 cell lines

Drug-resistant cell lines were isolated from loss-of-function KBM7 screens from wells containing single colonies. Cell lines with retroviral insertions in genes of interest were identified by screening for altered gene expression by RT-qPCR (see below). To ensure clonality of the gene-trapped cell lines for subsequent experiments, single colonies were isolated by serial dilution and propagated in standard growth medium. The location of the retroviral insertion sites in the clonal cell lines was identified using a similar strategy as in haploid screens except that the PCR products were sequenced by Sanger sequencing using primer CCseq.

## RT-qPCR assay

HEK293T or KBM7 cells were grown in 24-well plates and total RNA was extracted from 2 million cells using the RNeasy kit (Qiagen). cDNA was synthesized from 0.5 μg total RNA using Superscript III reverse transcriptase (Invitrogen, Carlsbad, CA) and oligo(dT)$_{20}$ primers (Invitrogen), following the manufacturer's instructions. Reactions were diluted two-fold with H$_2$O and 10 μL qPCR reactions were set up using 2× SYBR Green PCR Master mix (Thermo Fisher Scientific), 2 μL diluted cDNA preparation, and 0.2 μM of primers. Reactions were monitored using the Stratagene MX3000P qPCR system (Agilent, Santa Clara, CA). Primer pairs CC085/CC086 and CC097/CC098 were used for quantification of *ETNK1* expression levels. Primer pairs CC089/CC090 and CC133/CC134 were used for quantification of *PCYT2* expression levels. Primer pairs CC042/CC043 and CC044/CC045 were used for quantification of *ABCG2* expression levels. Primer pair GAPDH7-8f/GAPDH7-8r was used for quantification of *GAPDH* expression levels. Relative expression levels were quantified by the $\triangle\triangle C_T$ method using *GAPDH* as reference gene. Values reported are the average $\triangle\triangle C_T$ values calculated with the two primer pairs used. All measurements were performed in triplicate for each primer pair.

## Cell viability measurement

For KBM7 cell lines, 2000 cells were seeded per well in a 96-well plate in 90 μL growth medium. OPA was diluted in 10 μL growth medium and added to wells. After 72 hr incubation at 37°C and 5% CO$_2$, 100 μL of CellTiter-Glo reagent (Promega, Madison, WI) was added to each well. After homogenization, 100 μL of the resulting solution was transferred to a black opaque 96-well plate and luminescence was recorded on a Perkin Elmer (Waltham, MA) TopCount NXT system.

For HEK293T cell lines, cultures were grown to 80–90% confluence. Cell lines were subsequently seeded in wells of a 96-well plate at a dilution of 1:75 in 90 μL medium. After 16–18 hr incubation at 37°C and 5% CO$_2$, the remaining steps of the assay were performed as described above.

## Quantification of PE levels

Cells were washed with Tris buffer saline (TBS) (20 mM Tris, 150 mM NaCl, pH 7.6) and total lipids were extracted using the Folch method (Folch et al., 1957). Briefly, cells were resuspended in 20 vol of $CHCl_3$/MeOH 2:1 (v/v), homogenized for 20 min at room temperature and extracted using 4 vol of NaCl 0.9%. After drying the lower phase using a stream of nitrogen, lipids were resuspended in 2 vol of $CHCl_3$/MeOH 2:1 (v/v) and separated on Silica Gel 60 thin layer chromatography (TLC) plates (EMD Millipore, Darmstadt, Germany) according to the method of Skipski et al. (Skipski et al., 1964). Plates were developed using $CHCl_3$/MeOH/AcOH/$H_2O$ (50:30:8:3, v/v/v/v). Phospholipids were visualized by iodine staining and their identity was determined using standards. Phospholipids were then quantified by phosphate determination on scraped silica gel spots using the 'micro' assay as described by Zhou and Arthur (1992). For each experiment, the spot corresponding to PE was scraped into a tube. All other spots in the lane visualized by iodine staining were scraped into a second tube. The quantity of phosphate ($P_i$) in each sample was determined using standards consisting of known amounts of $KH_2PO_4$. The relative cellular PE content was estimated using the following formula: PE content = Quantity of $P_i$ in PE tube/(Quantity of $P_i$ in PE tube + Quantity of $P_i$ in tube containing all other phospholipids).

## Complementation of KBM7 cell lines

PCYT2 was re-expressed in PCYT2[GT] cells using the lentiviral vector pLJM1 (Sancak et al., 2008). cDNA encoding PCYT2 was amplified by PCR from cDNA prepared from total HEK293T RNA using primers CC109 and CC110. The plasmid Flag pLJM1 RagB wt (Addgene #19313) was digested with SalI and EcoRV. Amplified PCYT2 cDNA was inserted into pLJM1 by Gibson assembly and the constructed plasmid propagated in stbl2 cells (Invitrogen). Plasmid pLJM1-EGFP (Addgene #19319) was used as a negative control.

Lentivirus was produced in HEK293T cells using pLJM1-based vectors together with packaging vector psPAX2 (Addgene #12260) and envelope vector pCMV-VSV-G (Addgene #8454). KBM7 cell lines were infected with lentiviral particles in the presence of 8 µg/mL polybrene and selected in 0.4 µg/mL puromycin for 6 days. Transduced cell lines were maintained in growth medium supplemented with 0.3 µg/mL puromycin during subsequent assays.

## Knockdown of *PCYT2* in HEK293T cells

Constructs expressing shRNAs targeting PCYT2 were based on pLKO.1-TRC (Addgene #10878) (Moffat et al., 2006). Five independent shRNAs (TRCN0000236037, TRCN0000035648, TRCN0000236039, TRCN0000236038, TRCN0000236040) designed by the RNAi consortium (Broad Institute) were used to construct HEK293T knockdown cell lines as described by Addgene. Briefly, shRNA oligos were cloned into pLKO.1-TRC and the constructed plasmids were transfected into HEK293T cells together with psPAX2 and pCMV-VSV-G to produce lentivirus. HEK293T cells were then infected with lentiviral particles and selected in 2.0 µg/mL puromycin for 6 days. Cell lines stably expressing shRNAs of interest were maintained in growth medium supplemented with 1.5 µg/mL puromycin during subsequent assays. Scrambled shRNA in pLKO.1 (Addgene #1864) and empty vector pLKO.1-TRC were used as negative controls. The two shRNA sequences that achieved the highest knockdown efficiency were: kd1 (TCACGGCAAGACAGAATTAT, TRCN0000035648) and kd2 (ACTAGAGACCCTGGACAAATA, TRCN0000236039).

## Kennedy pathway activity measurements

HEK293T and HCT116 cells were grown to 30–40% confluence in 100 mm dishes in 20 mL medium. OPA or meclizine dihydrochloride (>97%) was diluted to 500 µL in growth medium and added to the dishes. After 5 hr incubation at 37°C and 5% $CO_2$, 0.5 µCi ethanolamine [1,2-[14]C] was added to each dish and cells were incubated for an additional 24 hr. Cellular PE content was quantified as described above. For determination of [14]C]-PE levels, phospholipids were separated by TLC as described above. After iodine staining, silica spots corresponding to PE were scraped into scintillation vials containing 5 mL of Ready-Solv HP scintillation cocktail (Beckman Coulter, Brea, CA) and [14]C] counts per minute were measured on a Beckman Coulter LS 6500 system. Levels of [14]C]-PE were normalized to the total amount of phospholipids in each sample determined by phosphate determination.

## Cell viability assay in presence of exogenous molecules

This assay was performed as the cell viability assay described above with the following modifications. HEK293T cells were seeded in 96-well plates at a dilution of 1:75 in 80 µL medium. Phospholipids were prepared from 5 or 10 mg/mL stocks in CHCl$_3$ and were first diluted fivefold in MeOH and then fivefold in growth medium. Organic solvents were degassed at 37°C and 10 µL was added to the wells. OPA was diluted in 10 µL growth medium and added to the wells. The rest of the assay was performed as above. Ethanolamine, triethanolamine, O-phosphorylethanolamine, and serine were prepared as 1 M solutions in H$_2$O, adjusted to pH 7 with HCl or NaOH, and diluted with H$_2$O in order to prepare 20 stock solutions. These solutions were diluted to 10 µL with growth medium and added to the wells. For experiments with scavengers of 1,4-dicarbonyls, lysine·HCl was prepared as a 1 M stock in H$_2$O and adjusted to pH 7 with NaOH, and salicylamine and pentyl-pyridoxamine were prepared as 50 mM stocks in H$_2$O. All solutions were diluted with H$_2$O in order to prepare 20x stock solutions and used as above.

## In vitro reactions of OPA with ethanolamine

OPA was adjusted to 100 µM in 100 µL PBS and to a final concentration of 2% DMSO. Then 10 µL of either neat ethanolamine (or ethanol as control) was added. After 2 hr at 37°C, OPA was extracted using 400 µL of CHCl$_3$/MeOH 2:1 (v/v). The lower phase was dried using a stream of nitrogen and resuspended in 100 µL ethanol. The recovery of OPA was assumed to be 100% and the reactions were tested in cell viability assays.

For LC-MS/MS analysis, OPA was adjusted to 0.5 mM in 50 µL H$_2$O (4% DMSO) and 3 µL of neat ethanolamine was added to the solution. For the 'ethanolamine only' control, OPA was replaced by DMSO. For the 'OPA only' control, no ethanolamine was added. After 2 hr at 37°C, the reactions were diluted 50-fold in MeOH before LC-MS/MS analysis.

## LC-MS/MS analysis

LC-MS/MS analysis was performed on a Thermo q-Exactive Plus mass spectrometer coupled to a Thermo Ultimate 3000 uHPLC (Thermo Fisher Scientific). The HPLC method used a Phenomenex (Torrance, CA) Kinetex C18 column (2.6 µm particle size, 10 nm pore size, 150 mm length, and 2.1 mm internal diameter) at a constant flow rate of 0.2 mL/min. Mobile phase A was 0.1% formic acid in H$_2$O (v/v) and mobile phase B was 0.1% formic acid in CH$_3$CN (v/v). A 10 µL sample was injected onto the column at 0% B and washed at this solvent composition for 3 min. The gradient was first increased to 10% B in 0.1 min and then to 100% B over the next 26.9 min. Detection on the q-Exactive Plus mass spectrometer was performed in positive mode between 300 and 2000 m/z, using an acquisition target of 3E6, and a maximum ion injection time of 200 ms at a resolution of 70,000 for MS and 35,000 for MS/MS data. For MS/MS experiments, [M + H]$^+$ ions were targeted for isolation and fragmentation at a normalized collision energy of 35 eV.

## Ehrlich's test

A 10 µL sample of OPA 5 mM in DMSO and 5 µL of ethanolamine 1 M (pH 7, prepared above) were added to 85 µL PBS. Control reactions were prepared by either replacing OPA by DMSO or ethanolamine by triethanolamine. After 3 hr at 37°C, OPA was extracted with CHCl$_3$/MeOH 2:1 (v/v). The lower phase was dried and resuspended in 35 µL ethanol. Ehrlich's reagent was prepared according to *Amarnath et al. (2004)* (80 mM 4-(dimethylamino)benzaldehyde in MeOH/0.6 M HCl 1:1 (v/v)). The resuspended reactions were diluted to 0.5 mL with H$_2$O and then to 1 mL with Ehrlich's reagent. After 2 min at 68°C, the solutions were cooled, transferred to a quartz cuvette, and the absorbance was measured between 450 and 700 nm (SpectraMax Plus 384, Molecular Devices, Sunnyvale, CA).

## Detection of PE-OPA adducts in vitro

A 5.2 µL sample of OPA 5 mM in DMSO and 2 µL of transphosphatidylated chicken egg PE (13 mM in CHCl$_3$) were added to 44.8 µL reaction buffer (1 M triethylammonium acetate/CHCl$_3$/MeOH 1:1:3 (v/v/v), as described by *Sullivan et al. [2010]*). Control reactions were performed by replacing either OPA by DMSO, PE by CHCl$_3$, or PE by chicken egg PC (13 mM in CHCl$_3$). After 3 hr at 37°C, the reactions were diluted to 500 µL with MeOH and 50 µL of the diluted reactions was dried using a stream of nitrogen. After resuspension in 25 µL MeOH, 225 µL PBS was added and the suspension

was sonicated in a water bath for 2 min. Then 5 μL (275 U) phospholipase D (PLD) from *Streptomyces chromofuscus* (Enzo Life Sciences) was added and after 16 hr at 37°C, the reaction was extracted with 1 mL $CHCl_3$/MeOH 2:1 (v/v). The lower phase was dried, resuspended in 1 mL $CHCl_3$, and analyzed by LC-MS/MS.

### Detection of PE-OPA adducts in human cells

Cells were seeded at 10% confluence in 100 mm dishes and were grown to 60% confluence in standard conditions. OPA was added to 250 nM for HEK293T and 450 nM for HCT116 in 15 mL complete growth medium. After 24 hr at 37°C and 5% $CO_2$, the cells were washed with 15 mL TBS and then resuspended in 20 vol of $CHCl_3$/MeOH 2:1 (v/v) and 0.6 vol of pentyl-pyridoxamine 50 mM. After homogenization for 20 min at room temperature, the suspension was extracted using 4 vol of NaCl 0.9%. The lower phase was dried using a stream of nitrogen and resuspended in 100 μL MeOH by sonication. A 50 μL sample of the suspension was diluted to 500 μL in PBS and further sonicated. Then 15 μL PLD (825 U) was added and the suspension was incubated at 37°C for 14 hr. The reaction was extracted with 2 mL $CHCl_3$/MeOH 2:1 (v/v) and the lower phase was dried and resuspended in 0.5 mL $CHCl_3$/MeOH 2:1 (v/v). A second extraction was performed by addition of 125 μL NaCl 0.9%. The lower phase was dried, resuspended in 180 μL $CHCl_3$, and analyzed by LC-MS/MS.

### Preparation of large unilamellar vesicles (LUVs)

Solutions containing a total of 1 mg of dioleoyl-PE and dioleoyl-PC (Avanti Lipids) were prepared from 10 mg/mL stock solutions in $CHCl_3$. The solvent was removed from these solutions using first a stream of nitrogen and then by evaporation under vacuum. Lipid films were rehydrated at 37°C for 5 hr with 0.4 mL 100 mM calcein pH 7.4, and homogenized by vortexing and five freeze-thaw cycles. Liposome suspensions were next extruded 20 times through 100 nm polycarbonate filters (Avanti Mini-Extruder) to generate LUVs encapsulating calcein. LUVs were separated from free calcein by gel filtration over Sephadex G-50 using 20 mM HEPES pH 7.5, 150 mM NaCl, 1 mM EDTA as buffer. Leakage experiments were started immediately after chromatography.

### Liposome leakage assay

A 50 μL sample of calcein-containing LUVs diluted in 20 mM HEPES pH 7.5, 150 mM NaCl, 1 mM EDTA was dispensed in 96-well plates. Then 50 μL OPA solution (diluted from 200 stocks in the same buffer as the LUVs) was added to start the leakage assay. The fluorescence of calcein was monitored each 30 s on a Spectramax i3 (Molecular Devices) using excitation at 493 nm and emission at 518 nm over 65 min and at room temperature. Background fluorescence was subtracted for each OPA concentration by using control wells in which calcein-containing LUVs were replaced by buffer. A control using DMSO vehicle instead of OPA was used as the 0% leakage reference and a solution of 0.2% Triton X-100 was used instead of the OPA solution in the 100% leakage reference. For each time point, the fluorescence data were normalized to these two reference samples.

## Acknowledgements

We acknowledge D Sabatini (Whitehead Institute) for providing mutagenized KBM7 cells and V Amarnath (Vanderbilt University) for providing reagents and advice. We thank V Vijayan, K Amarnath, T Peterson, and members of the O'Shea lab for advice and assistance. We thank A Darnell for critical reading of the manuscript.

## Additional information

### Competing interests

EKO: Chief Scientific Officer and a Vice President at the Howard Hughes Medical Institute, one of the three founding funders of *eLife*. The other authors declare that no competing interests exist.

## Funding

| Funder | Grant reference number | Author |
| --- | --- | --- |
| Howard Hughes Medical Institute | | Christopher Chidley Erin K O'Shea |
| Novartis Foundation | | Christopher Chidley |
| Schweizerischer Nationalfonds zur Förderung der Wissenschaftlichen Forschung | PBELP3-135869 | Christopher Chidley |

The funders had no role in study design, data collection and interpretation, or the decision to submit the work for publication.

## Author contributions

CC, performed experiments, analyzed and interpreted data, performed and analyzed LC-MS measurements, designed experiments, wrote the manuscript; SAT, performed and analyzed LC-MS measurements; KB, prepared mutagenized KBM7 cells; EKO, designed experiments, wrote the manuscript

## Author ORCIDs

Christopher Chidley, http://orcid.org/0000-0002-8212-3148
Erin K O'Shea, http://orcid.org/0000-0002-2649-1018

# Additional files

## Supplementary files

• Supplementary file 1. Oligonucleotides used in this study.

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
