## [Decision Letter]

Thank you for submitting your work entitled "The anticancer natural product ophiobolin A induces cytotoxicity by covalent modification of phosphatidylethanolamine" for consideration by *eLife*. Your article has been favorably evaluated by Michael Marletta (Senior editor) and three reviewers, one of whom is a member of our Board of Reviewing Editors.

The reviewers have discussed the reviews with one another and the Reviewing Editor has drafted this decision to help you prepare a revised submission.

This study investigates the basis of ophiobolin A (OPA) cytotoxicity using a genetic screening strategy and followed up with a multi-faceted analysis to reach the surprising conclusion that OPA kills cancer cells by covalently modifying phosphatidylethanolamine (PE). The haploid insertion strategy outlined here is a powerful way to screen for small molecule targets by looking for gene inactivating mutants conferring resistance and it was nicely validated with topoisomerase inhibitors. It was also impressive that the three inactivated genes uncovered in promoting OPA resistance were all part of the PE biosynthetic pathway. Based on the literature and the tendency of OPA to undergo pyrrole formation with primary amines, the authors astutely postulated that PE could be interacting directly with OPA, and this was in fact observed. The authors also found that knockdown of the PE biosynthetic enzymes recapitulated the genetic screen findings about OPA resistance and that PE added exogenously could confer OPA resistance, presumably by titrating OPA out. Overall, this is a fascinating study with a very unusual proposed mechanism of action of a cytotoxic drug. However, it is believed that more compelling evidence is needed to support the conclusion that the PE reaction with OPA is the primary mechanism of action for OPA's cell killing.

Requested revisions:

It is somewhat surprising that these modest changes in the levels of PE (16-24%) in the genetic experiments could be so important for the influencing potency of the compound in light of the model for OPA action. In addition, the dose response curve in Figure 1—figure supplement 4, shows only a very modest effect on OPA IC_50_, and this also leads to questions about the mechanism of action of OPA. It raises the concern that there still is one or more Lys containing protein targets that are of great importance for OPA pharmacology as the authors state has been suggested before. Loss of function screens may have difficulty in revealing the major target of antiproliferative small molecules such as OPA, because inactivation of the very target is expected to recapitulate the cellular phenotype of OPA and the mutant cells would not be viable.

A significant concern relates to a paper that they cite by Dasari et al. (2015) that notes that the 6-epi isomer of OPA is 40-fold less potent than the natural OPA across a panel of cell lines. Since the pyrrole OPA adduct would be the same regardless of the starting stereoisomer of OPA (as this 6-stereocenter is lost as it becomes sp2), one would then have to argue that the rate of pyrrole formation between PE and the 6-epimer is 40-fold slower than the reaction with the natural compound. While not impossible, this large difference in reactivity seems unlikely as there would not appear to be complex recognition involved in the two small molecules coming together. Nevertheless, the PE reactivity mechanistic model should be readily tested by getting some of the 6-epimer from Dasari et al., or some other supplier. If the chemical reactivity of the 6-epimer with PE is indeed slowed by ca. 40-fold as the above analysis predicts, this would be very reassuring for the authors' model. This model could also be tested by comparing a series of other analogs from Dasari et al. (or other suppliers) and determining the correlation between cell potency and PE reactivity.

If the PE-OPA adduct is indeed responsible for the cytotoxicy of OPA, one might have expected that the addition of exogenous PE could enhance, rather than reduce, the cytotoxicity of OPA. Perhaps the PE-OPA adduct needs to be embedded in the membrane to be toxic. It would be helpful if the authors could perform preparative quantities of the PE-OPA adduct and then add this adduct exogenously to the cell culture medium to explore its potential to be toxic to cell growth.

It is possible that the PE-OPA adduct has unique fluorescence compared with the PE and OPA as isolated molecules. If so, the authors could check to see if OPA addition induces the corresponding fluorescence change that might be detected by cellular imaging. Alternatively, perhaps the authors could check to see if OPA drives leakiness in the cell membrane by allowing normally cell impenetrant colorimetric dyes to accumulate in cells rapidly after OPA administration.

---

## [Author Response]

Requested revisions:

It is somewhat surprising that these modest changes in the levels of PE (16-24%) in the genetic experiments could be so important for the influencing potency of the compound in light of the model for OPA action. In addition, the dose response curve in Figure 1—figure supplement 4, shows only a very modest effect on OPA IC_50_, and this also leads to questions about the mechanism of action of OPA. It raises the concern that there still is one or more Lys containing protein targets that are of great importance for OPA pharmacology as the authors state has been suggested before. Loss of function screens may have difficulty in revealing the major target of antiproliferative small molecules such as OPA, because inactivation of the very target is expected to recapitulate the cellular phenotype of OPA and the mutant cells would not be viable.

While we agree that loss-of-function screens typically cannot be used to directly identify genes that encode targets essential for cell viability, we believe that the results of the haploid genetic screen we performed provide strong evidence that PE is the main target of OPA. We found that genetic inactivation of the Kennedy pathway in KBM7 cells leads to a reduction in total cellular PE levels and a concomitant increase in OPA resistance. We also found that OPA reacts covalently with PE in human cells, leading us to hypothesize that the observed cytotoxicity of OPA reflects the amount of covalent PE-OPA adducts formed in cells.

Although the Kennedy pathway is genetically inactivated in the mutant cells detected as OPA-resistant in our screen, these cells are not devoid of PE, as PE can be synthesized via multiple pathways and cells employ mechanisms to maintain phospholipid homeostasis. As such, these mutant cells have only a 16-24% reduction in PE levels (Figure 1), and any phenotype associated with this reduction is therefore likely to be mild. The modest reduction in PE levels was accompanied by a modest increase in the IC_50_ of OPA from 43 nM in wild-type KBM7 cells to 70-85 nM in the mutants (Figure 1—figure supplement 4). Based on the mechanism of the Paal-Knorr reaction, we surmise that the rate of covalent adduct formation is proportional to both the concentration of OPA and that of PE in the lipid bilayer. Accordingly, a modest reduction in PE levels (less than 2-fold reduction) translates into a modest decrease in the rate of deleterious adduct formation and thus a modest increase in OPA resistance (up to 2-fold). This slight increase in resistance manifests as a quantitatively detectable increase in proliferation rate of the mutants’ cells upon OPA treatment in a sensitive, sequencing-based genetic screen output. We used these subtle changes found in the mutant cells to form a hypothesis about the identity of the direct molecular target of OPA in human cells.

The reviewers raise the concern that "there still is one or more Lys containing protein targets that are of great importance for OPA pharmacology as the authors state has been suggested before". If lysine-containing protein targets were the major determinants of OPA cytotoxicity rather than PE-OPA adduct formation, a mild decrease in PE content would likely have little or no effect on the resistance to OPA. The results obtained in the haploid genetic screen and the follow-up experiments are fully and quantitatively accounted for by our model of OPA cytotoxicity. We therefore believe that PE-OPA adduct formation is the main determinant of OPA cytotoxicity. We have edited the manuscript to clarify our interpretation of the genetic experiments.

A significant concern relates to a paper that they cite by Dasari et al. (2015) that notes that the 6-epi isomer of OPA is 40-fold less potent than the natural OPA across a panel of cell lines. Since the pyrrole OPA adduct would be the same regardless of the starting stereoisomer of OPA (as this 6-stereocenter is lost as it becomes sp2), one would then have to argue that the rate of pyrrole formation between PE and the 6-epimer is 40-fold slower than the reaction with the natural compound. While not impossible, this large difference in reactivity seems unlikely as there would not appear to be complex recognition involved in the two small molecules coming together. Nevertheless, the PE reactivity mechanistic model should be readily tested by getting some of the 6-epimer from Dasari et al., or some other supplier. If the chemical reactivity of the 6-epimer with PE is indeed slowed by ca. 40-fold as the above analysis predicts, this would be very reassuring for the authors' model. This model could also be tested by comparing a series of other analogs from Dasari et al. (or other suppliers) and determining the correlation between cell potency and PE reactivity.

As pointed out by the reviewers, Dasari et al. observe a significant difference in potency between 6-*epi*-OPA and OPA across a panel of human cell lines despite the fact that both compounds would form the same pyrrole adduct upon reaction with PE (Dasari et al., 2015). Given our model that cytotoxicity depends on the amounts of PE-OPA adduct formed, this observation must mean that the reaction of PE with 6-*epi-*OPA is on the order of 40-fold slower than the reaction of PE with OPA. The reviewers suggest that this large difference in reactivity is improbable as the initial step in the reaction (hemiaminal formation) is unlikely to be influenced by the stereochemistry of OPA at position 6.

Although we agree with this point, the difference in reactivity between the OPA epimers can readily be explained by considering the rate-limiting step of this reaction. Amarnath et al. observed that the rate limiting step in the Paal-Knorr reaction is the ring cyclization step and not hemiaminal formation (Amarnath et al., 1991). Amarnath et al. also observed that the cyclization step is indeed influenced by the stereochemistry of the 1,4 diketone, and rates of pyrrole formation can be up to 57-fold different between stereoisomers (Amarnath et al., 1991). In light of this careful documentation of the effect of stereochemistry on the rate-limiting step in the Paal-Knorr reaction, the 40-fold difference in potency between OPA and 6-*epi*-OPA can be attributed to a reduction in PE-OPA pyrrole adduct formation.

Lastly, 6-*epi*-OPA cannot be obtained from a commercial supplier, and thus we hope that this theoretical explanation will suffice to satisfy the reviewers regarding this interesting aspect of the Paal-Knorr reaction.

If the PE-OPA adduct is indeed responsible for the cytotoxicy of OPA, one might have expected that the addition of exogenous PE could enhance, rather than reduce, the cytotoxicity of OPA. Perhaps the PE-OPA adduct needs to be embedded in the membrane to be toxic. It would be helpful if the authors could perform preparative quantities of the PE-OPA adduct and then add this adduct exogenously to the cell culture medium to explore its potential to be toxic to cell growth.

The reviewers state that "one might have expected that the addition of exogenous PE could enhance, rather than reduce, the cytotoxicity of OPA". According to our model, the addition of exogenous PE could indeed lead to an increase in cytotoxicity if it results in an increase in the amount of cytotoxic PE-OPA adducts in the cell membrane. However, this result would only be obtained in hypothetical scenarios such as: (1) exogenous PE can efficiently partition to the cell membrane when added to aqueous culture medium and the increase in lipid bilayer PE content leads to an increase in PE-OPA adduct formation; or (2) covalent PE-OPA adducts formed in the cell culture medium can partition efficiently into the lipid bilayer in a manner which retains their cytotoxic potential.

Our experiments clearly show that addition of exogenous PE to the growth medium of HEK293T cells reduces the cytotoxicity of OPA (Figure 3 and Figure 3—figure supplement 1). To explain this data, we surmise that OPA reacts with the exogenously added PE in the growth medium and that the large majority of these adducts do not partition into the cell membrane, thereby reducing the number of OPA molecules available to partition into the cell membrane, react with endogenous PE, and induce cell death. Supporting this assertion is the fact that the solubility of PE in aqueous solutions is very low, and thus exogenous PE forms visible insoluble aggregates when added to cell culture medium. Thus, any OPA reacting with exogenously added PE will likely remain trapped in these aggregates as the PE-OPA adducts are themselves also very hydrophobic.

We edited the manuscript to clarify our prediction of the effect of exogenous PE addition to cells treated with OPA.

The reviewers suggest that we generate "preparative quantities of the PE-OPA adduct and then add this adduct exogenously to the cell culture medium to explore its potential to be toxic to cell growth". This experiment is conceptually an outstanding way of proving the role of PE-OPA adducts in the observed cytotoxicity of OPA and would provide a strong validation of our mechanism of action. However, this experiment is technically challenging as hydrophobic PE-OPA adducts need to be delivered to cells in sufficient amounts, and their final membrane orientation must mimic that of endogenously generated adducts.

We attempted the suggested adduct delivery experiment using two approaches. We first synthesized preparative amounts of PE-OPA adduct and dissolved the synthesized molecule in DMSO. We then added the synthesized adduct to HEK293T cells at final concentrations of up to 100 μM but did not observe any cytotoxicity. As mentioned earlier, we believe that this is because the PE-OPA adduct is extremely hydrophobic and its solubility in aqueous solutions is accordingly low. We believe that upon addition to cell culture medium, the adduct forms insoluble aggregates and cannot partition into the cell membrane to a sufficient degree to cause toxicity.

In order to circumvent the problem of inefficient delivery of hydrophobic molecules to animal cells, we next attempted liposomal delivery of the adduct. We incorporated pre-formed PE-OPA adduct into fusogenic liposomes using Ibidi's Fuse-it-L commercial reagent and added the prepared liposomes to HEK293T cells, but were also not able to observe significant cytotoxicity. We believe this can be explained by the difficulty in delivering sufficient quantities of adduct using this system which is typically used to deliver analytical quantities of fluorescent probes. This difficulty is likely exacerbated by the strong lipid bilayer destabilization effect of the adduct as observed in the liposome leakage assay described above – any liposomes with significant quantities of PE-OPA adduct would likely be structurally unstable, perhaps impairing or reducing fusion with the cell membrane.

It is possible that the PE-OPA adduct has unique fluorescence compared with the PE and OPA as isolated molecules. If so, the authors could check to see if OPA addition induces the corresponding fluorescence change that might be detected by cellular imaging. Alternatively, perhaps the authors could check to see if OPA drives leakiness in the cell membrane by allowing normally cell impenetrant colorimetric dyes to accumulate in cells rapidly after OPA administration.

Unlike PE and OPA, the PE-OPA adduct contains a pyrrole moiety that weakly absorbs in the near ultraviolet range of the electromagnetic spectrum. However, this weak absorption lies in a spectral region that is subject to significant cellular auto-fluorescence, and therefore is not suitable for fluorescence imaging. Additionally, it is unlikely that the fluorescence of the PE-OPA adduct will be distinguishable from that of OPA adducts formed by reaction with other primary amines, and therefore this experiment would not provide further evidence supporting PE as the primary molecular target of OPA.

According to our model of the mechanism of action of OPA, the formation of PE-OPA adducts leads to adverse structural effects on the lipid bilayer and ultimately to cell death. As suggested by the reviewers, we should therefore be able to detect cell membrane leakiness induced by OPA treatment. However, these experiments must be able to distinguish between cell membrane leakiness induced directly by PE-OPA adduct formation and leakiness due to downstream effects of cell death and lysis through an unrelated mechanism of action. In addition, it was important for us to prove that leakiness depends not only on OPA concentration but also on PE levels in the lipid bilayer, as this must be true if the increase in resistance to OPA we observed in Kennedy pathway mutants is due to reduced PE levels.

To address these points, we tested if OPA induces membrane leakiness in vitro using artificial liposomes. We prepared liposomes loaded with the fluorescent dye calcein using a variable ratio of dioleoyl-PE to dioleoyl-PC. As calcein fluorescence becomes dequenched upon release from liposomes and resulting dilution, an increase in the fluorescence of the aqueous solution containing liposomes can be used as an indicator of liposome leakage. Excitingly, we observed that OPA treatment caused extensive liposome leakage, and that the leakage was directly proportional to the PE content of the liposomes (Figure 5). Importantly, we observed that liposomes composed only of PC or with very low PE content remain intact even at high OPA concentration, indicating that the leakage is likely specifically due to the presence of PE-OPA adducts. These results clearly show that OPA destabilizes lipid bilayers and thereby support our conclusion that formation of PE-OPA adducts is a primary mechanism of cytotoxicity.